# Evolving Reinforcement Learning Algorithms

**John D. Co-Reyes, Yingjie Miao, Daiyi Peng, Esteban Real,**
**Sergey Levine**, **Quoc V. Le**, **Honglak Lee**, **Aleksandra Faust**[*]
Research at Google, Mountain View, CA 94043, USA

## ABSTRACT

We propose a method for meta-learning reinforcement learning algorithms by searching over the space of computational graphs which compute the loss function for a value-based model-free RL agent to optimize. The learned algorithms are domain-agnostic and can generalize to new environments not seen during training. Our method can both learn from scratch and bootstrap off known existing algorithms, like DQN, enabling interpretable modifications which improve performance. Learning from scratch on simple classical control and gridworld tasks, our method rediscovers the temporal-difference (TD) algorithm. Bootstrapped from DQN, we highlight two learned algorithms which obtain good generalization performance over other classical control tasks, gridworld type tasks, and Atari games. The analysis of the learned algorithm behavior shows resemblance to recently proposed RL algorithms that address overestimation in value-based methods.

## 1 INTRODUCTION

Designing new deep reinforcement learning algorithms that can efficiently solve across a wide variety of problems generally requires a tremendous amount of manual effort. Learning to design reinforcement learning algorithms or even small sub-components of algorithms would help ease this burden and could result in better algorithms than researchers could design manually. Our work might then shift from designing these algorithms manually into designing the language and optimization methods for developing these algorithms automatically.

Reinforcement learning algorithms can be viewed as a procedure that maps an agent's experience to a policy that obtains high cumulative reward over the course of training. We formulate the problem of training an agent as one of meta-learning: an outer loop searches over the space of computational graphs or programs that compute the objective function for the agent to minimize and an inner loop performs the updates using the learned loss function. The objective of the outer loop is to maximize the training return of the inner loop algorithm.

Our learned loss function should generalize across many different environments, instead of being specific to a particular domain. Thus, we design a search language based on genetic programming (Koza, 1993) that can express general symbolic loss functions which can be applied to any environment. Data typing and a generic interface to variables in the MDP allow the learned program to be domain agnostic. This language also supports the use of neural network modules as subcomponents of the program, so that more complex neural network architectures can be realized. Efficiently searching over the space of useful programs is generally difficult. For the outer loop optimization, we use regularized evolution (Real et al., 2019), a recent variant of classic evolutionary algorithms that employ tournament selection (Goldberg & Deb, 1991). This approach can scale with the number of compute nodes and has been shown to work for designing algorithms for supervised learning (Real et al., 2020). We adapt this method to automatically design algorithms for reinforcement learning.

While learning from scratch is generally less biased, encoding existing human knowledge into the learning process can speed up the optimization and also make the learned algorithm more interpretable. Because our search language expresses algorithms as a generalized computation graph, we

---

[*]jcoreyes@eecs.berkeley.edu, {yingjiemiao,daiyip,ereal,slevine,qvl,honglak,sandrafaust}@google.com

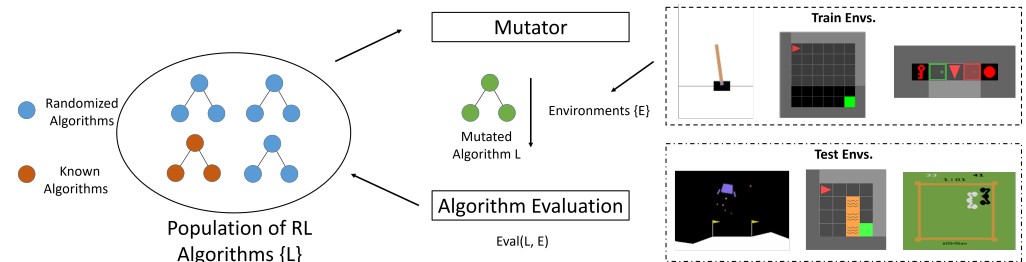

Figure 1: Method overview. We use regularized evolution to evolve a population of RL algorithms. A mutator alters top performing algorithms to produce a new algorithm. The performance of the algorithm is evaluated over a set of training environments and the population is updated. Our method can incorporating existing knowledge by starting the population from known RL algorithms instead of purely from scratch.

can embed known RL algorithms in the graphs of the starting population of programs. We compare starting from scratch with bootstrapping off existing algorithms and find that while starting from scratch can learn existing algorithms, starting from existing knowledge leads to new RL algorithms which can outperform the initial programs.

We learn two new RL algorithms which outperform existing algorithms in both sample efficiency and final performance on the training and test environments. The learned algorithms are domain agnostic and generalize to new environments. Importantly, the training environments consist of a suite of discrete action classical control tasks and gridworld style environments while the test environments include Atari games and are unlike anything seen during training.

The contribution of this paper is a method for searching over the space of RL algorithms, which we instantiate by developing a formal language that describes a broad class of value-based model-free reinforcement learning methods. Our search language enables us to embed existing algorithms into the starting graphs which leads to faster learning and interpretable algorithms. We highlight two learned algorithms which generalize to completely new environments. Our analysis of the meta-learned programs shows that our method automatically discovers algorithms that share structure to recently proposed RL innovations, and empirically attain better performance than deep Q-learning methods.

## 2 RELATED WORK

***Learning to learn*** is an established idea in in supervised learning, including meta-learning with genetic programming (Schmidhuber, 1987; Holland, 1975; Koza, 1993), learning a neural network update rule (Bengio et al., 1991), and self modifying RNNs (Schmidhuber, 1993). Genetic programming has been used to find new loss functions (Bengio et al., 1994; Trujillo & Olague, 2006). More recently, AutoML (Hutter et al., 2018) aims to automate the machine learning training process. Automated neural network architecture search (Stanley & Miikkulainen, 2002; Real et al., 2017; 2019; Liu et al., 2017; Zoph & Le, 2016; Elsken et al., 2018; Pham et al., 2018) has made large improvements in image classification. Instead of learning the architecture, AutoML-Zero (Real et al., 2020) learns the algorithm from scratch using basic mathematical operations. Our work shares similar ideas, but is applied to the RL setting and assumes additional primitives such as neural network modules. In contrast to AutoML-Zero, we learn computational graphs with the goal of automating RL algorithm design. Our learned RL algorithms generalize to new problems, not seen in training.

***Automating RL.*** While RL is used for AutoML (Zoph & Le, 2016; Zoph et al., 2018; Cai et al., 2018; Bello et al., 2017), automating RL itself has been somewhat limited. RL requires different design choices compared to supervised learning, including the formulation of reward and policy update rules. All of which affect learning and performance, and are usually chosen through trial and error. AutoRL addresses the gap by applying the AutoML framework from supervised learning to the MDP setting in RL. For example, evolutionary algorithms are used to mutate the value or actor network weights (Whiteson & Stone, 2006; Khadka & Tumer, 2018), learn task reward (Faust et al., 2019), tune hyperparameters (Tang & Choromanski, 2020; Franke et al., 2020), or search for a neural network architecture (Song et al., 2020; Franke et al., 2020). This paper focuses on task-agnostic RL update rules in the value-based RL setting which are both interpretable and generalizable.

***Meta-learning in RL.*** Recent work has focused on few-shot task adaptation. Finn et al. (2017); Finn & Levine (2018) meta-learns initial parameters which can quickly adapt to new tasks, while $\text{RL}^2$ (Duan et al., 2016) and concurrent work (Wang et al., 2017), formulates RL itself as a learning problem that is learned with an RNN. The meta-learned component of these works is tuned to a particular domain or environment, in the form of NN weights which cannot be used for completely new domains with potentially different sized inputs. Neural Programmer-Interpreters (Reed & De Freitas, 2015; Pierrot et al., 2019) overcome the environment generalization challenge by learning hierarchical neural programs with domain-specific encoders for different environments. Here, the computational graph has a flexible architecture and generalizes across different environments.

***Learning RL algorithms*** or their components, such as a reward bonus or value update function, has been studied previously with meta-gradients (Kirsch et al., 2020; Chebotar et al., 2019; Oh et al., 2020), evolutionary strategies (Houthooft et al., 2018), and RNNs (Duan et al., 2016). Although our work also learns RL algorithms, the update rule is represented as a computation graph which includes both neural network modules and symbolic operators. One key benefit is that the resulting graph can be interpreted analytically and can optionally be initialized from known existing algorithms. Prior work that focuses on learning RL losses, generalizes to different goals and initial conditions *within* a single environment (Houthooft et al., 2018), or learns a domain invariant policy update rule that can generalize to new environments (Kirsch et al., 2020). Another approach searches over the space of curiosity programs using a similar language of DAGs with neural network modules (Alet et al., 2020a) and performs the meta-training on a single environment. In contrast, our method is applied to learn general RL update rules and meta-trained over a diverse set of environments.

## 3    LEARNING REINFORCEMENT LEARNING ALGORITHMS

In this section, we first describe the problem setup. An inner loop method $\text{Eval}(L, \mathcal{E})$ evaluates a learned RL algorithm $L$ on a given environment $\mathcal{E}$. Given access to this procedure, the goal for the outer loop optimization is to learn a RL algorithm with high training return over a set of training environments. We then describe the search language which enables the learning of general loss functions and the outer loop method which can efficiently search over this space.

### 3.1    PROBLEM SETUP

We assume that the agent parameterized with policy $\pi_\theta(a_t|s_t)$ outputs actions $a_t$ at each time step to an environment $\mathcal{E}$ and receives reward $r_t$ and next state $s_{t+1}$. Since we are focusing on discrete action value-based RL methods, $\theta$ will be the parameters for a Q-value function and the policy is obtained from the Q-value function using an $\epsilon$-greedy strategy. The agent saves this stream of transitions $(s_t, s_{t+1}, a_t, r_t)$ to a replay buffer and continually updates the policy by minimizing a loss function $L(s_t, a_t, r_t, s_{t+1}, \theta, \gamma)$ over these transitions with gradient descent. Training will occur for a fixed number of $M$ training episodes where in each episode $m$, the agent earns episode return $R_m = \sum_{t=0}^{T} r_t$. The performance of an algorithm for a given environment is summarized by the normalized average training return, $\frac{1}{M} \sum_{m=1}^{M} \frac{R_i - R_{min}}{R_{max} - R_{min}}$, where $R_{min}$ and $R_{max}$ are the minimum and maximum re-

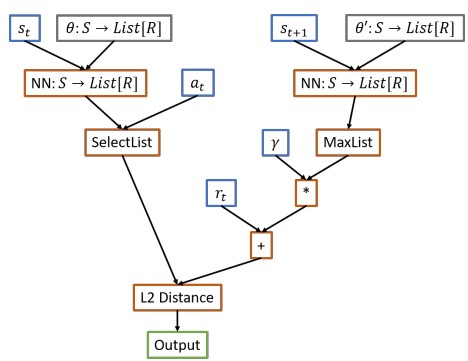

Figure 2: Visualization of a RL algorithm, DQN, as a computational graph which computes the loss $L = (Q(s_t, a_t) - (r_t + \gamma * \max_a Q_{targ}(s_{t+1}, a)))^2$. Input nodes are in blue, parameter nodes in gray, operation nodes in orange, and output in green.

turn for that environment. We assume these are known ahead of time. This inner loop evaluation procedure $\text{Eval}(L, \mathcal{E})$ is outlined in Algorithm 1. To score an algorithm, we use the normalized average training return instead of the final behavior policy return because the former metric will factor in sample efficiency as well.

The goal of the meta-learner is to find the optimal loss function $L(s_t, a_t, r_t, s_{t+1}, \theta, \gamma)$ to optimize $\pi_\theta$ with maximal normalized average training return over the set of training environments. The full objective for the meta-learner is:

$$L^* = \arg\max_L \left[ \sum_{\mathcal{E}} \text{Eval}(L, \mathcal{E}) \right]$$

$L$ is represented as a computational graph which we describe in the next section.

## 3.2 SEARCH LANGUAGE

Our search language for the algorithm $L$ should be expressive enough to represent existing algorithms while enabling the learning of new algorithms which can obtain good generalization performance across a wide range of environments. Similar to Alet et al. (2020a), we describe the RL algorithm as general programs with a domain specific language, but we target updates to the policy rather than reward bonuses for exploration. Algorithms will map transitions $(s_t, a_t, s_{t+1}, r_t)$, policy parameters $\theta$, and discount factor $\gamma$ into a scalar loss to be optimized with gradient descent. We express $L$ as a computational graph or directed acyclic graph (DAG) of nodes with typed inputs and outputs. See Figure 2 for a visualization of DQN expressed in this form. Nodes are of several types:

**Input nodes** represent inputs to the program, and include elements from transitions $(s_t, a_t, s_{t+1}, r_t)$ and constants, such as the discount factor $\gamma$.

**Parameter nodes** are neural network weights, which can map between various data types. For example, the weights for the Q-value network will map an input node with state data type to a list of real numbers for each action.

**Operation nodes** compute outputs given inputs from parent nodes. This includes applying parameter nodes, as well as basic math operators from linear algebra, probability, and statistics. A full list of operation nodes is provided in Appendix A. By default, we set the last node in the graph to compute the output of the program which is the scalar loss function to be optimized. Importantly, the inputs and outputs of nodes are typed among (state, action, vector, float, list, probability). This typing allows for programs to be applied to any domain. It also restricts the space of programs to ones with valid typing which reduces the search space.

---

**Algorithm 1** Algorithm Evaluation, $\text{Eval}(L, \mathcal{E})$

1: **Input:** RL Algorithm $L$, Environment $\mathcal{E}$, training episodes $M$
2: **Initialize:** Q-value parameters $\theta$, target parameters $\theta'$ empty replay buffer $\mathcal{D}$
3: **for** $i = 1$ to $M$ **do**
4:     **for** $t = 0$ **to** $T$ **do**
5:         With probability $\epsilon$, select a random action $a_t$,
6:         otherwise select $a_t = \arg\max_a Q(s_t, a)$
7:         Step environment $s_{t+1}, r_t \sim \mathcal{E}(a_t, s_t)$
8:         $\mathcal{D} \leftarrow \mathcal{D} \cup \{s_t, a_t, r_t, s_{t+1}\}$
9:         Update parameters $\theta \leftarrow \theta - \nabla_\theta L(s_t, a_t, r_t, s_{t+1}, \theta, \gamma)$
10:         Update target $\theta' \leftarrow \theta$
11:     **end for**
12:     Compute episode return $R_m = \sum_{t=0}^T r_t$
13: **end for**
14: **Output:**
15:     Normalized training performance $\frac{1}{M} \sum_{m=1}^M \frac{R_m - R_{min}}{R_{max} - R_{min}}$

**Algorithm 2** Evolving RL Algorithms

1: **Input:** Training environments $\{\mathcal{E}\}$, hurdle environment $\mathcal{E}_h$, hurdle threshold $\alpha$, optional existing algorithm $A$
2: **Initialize:** Population $P$ of RL algorithms $\{L\}$, history $H$, randomized inputs $I$. If bootstrapping, initialize $P$ with $A$.
3: Score each L in $P$ with $H[L].score \leftarrow \sum_{\mathcal{E}} \text{Eval}(L, \mathcal{E})$
4: **for** $c = 0$ to C **do**
5:     Sample tournament $T \sim Uniform(P)$
6:     Parent algorithm $L \leftarrow$ highest score algorithm in $T$
7:     Child algorithm $L' \leftarrow$ Mutate(L)
8:     $H[L'].hash \leftarrow \text{Hash}(L'(I))$
9:     **if** $H[L'].hash$ was new **and** $\text{Eval}(L', \mathcal{E}_h) > \alpha$ **then**
10:         $H[L'].score \leftarrow \sum_{\mathcal{E}} \text{Eval}(L', \mathcal{E})$
11:     **end if**
12:     Add $L'$ to population $P$
13:     Remove oldest $L$ from population
14: **end for**
15: **Output:** Algorithm L with highest score

---

## 3.3 EVOLUTIONARY SEARCH METHOD

Evaluating thousands of programs over a range of complex environments is prohibitively expensive, especially if done serially. We adapt a genetic programming (Koza, 1993) method for the search method and use regularized evolution (Real et al., 2019), a variant of classic evolutionary algorithms that employ tournament selection (Goldberg & Deb, 1991). Regularized evolution has been shown to work for learning supervised learning algorithms (Real et al., 2020) and can be parallelized across compute nodes. Tournament selection keeps a population of $P$ algorithms and improves the population through cycles. Each cycle picks a tournament of $T < P$ algorithms at random and selects the best algorithm in the tournament as a parent. The parent is mutated into a child algorithm which gets added to the population while the oldest algorithm in the population is removed. We use

a single type of mutation which first chooses which node in the graph to mutate and then replaces it with a random operation node with inputs drawn uniformly from all possible inputs.

There exists a combinatorially large number of graph configurations. Furthermore, evaluating a single graph, which means training the full inner loop RL algorithm, can take up a large amount of time compared to the supervised learning setting. Speeding up the search and avoiding needless computation are needed to make the problem more tractable. We extend regularized evolution with several techniques, detailed below, to make the optimization more efficient. The full training procedure is outlined in Algorithm 2.

**Functional equivalence check** (Real et al., 2020; Alet et al., 2020b). Before evaluating a program, we check if it is functionally equivalent to any previously evaluated program. This check is done by hashing the concatenated output of the program for 10 values of randomized inputs. If a mutated program is functionally equivalent to an older program, we still add it to the population, but use the saved score of the older program. Since some nodes of the graph do not always contribute to the output, parts of the mutated program may eventually contribute to a functionally different program.

**Early hurdles** (So et al., 2019). We want poor performing programs to terminate early so that we can avoid unneeded computation. We use the CartPole environment as an early hurdle environment $\mathcal{E}_h$ by training a program for a fixed number of episodes. If an algorithm performs poorly, then episodes will terminate in a short number of steps (as the pole falls rapidly) which quickly exhausts the number of training episodes. We use $\mathrm{Eval}(L, \mathcal{E}_h) < \alpha$ as the threshold for poor performance with $\alpha$ chosen empirically.

**Program checks.** We perform basic checks to rule out and skip training invalid programs. The loss function needs to be a scalar value so we check if the program output type is a float ($\mathrm{type}(L) = \mathbb{R}$). Additionally, we check if each program is differentiable with respect to the policy parameters by checking if a path exists in the graph between the output and the policy parameter node.

**Learning from Scratch and Bootstrapping.** Our method enables both learning from scratch and learning from existing knowledge by bootstrapping the initial algorithm population with existing algorithms. We learn algorithms from scratch by initializing the population of algorithms randomly. An algorithm is sampled by sampling each operation node sequentially in the DAG. For each node, an operation and valid inputs to that operation are sampled uniformly over all possible options.

While learning from scratch might uncover completely new algorithms that differ substantially from the existing methods, this method can take longer to converge to a reasonable algorithm. We would like to incorporate the knowledge we do have of good algorithms to bootstrap our search from a better starting point. We initialize our graph with the loss function of DQN (Mnih et al., 2013) so that the first 7 nodes represent the standard DQN loss, while the remaining nodes are initialized randomly. During regularized evolution, the nodes are not frozen, such that it is possible for the existing sub-graph to be completely replaced if a better solution is found.

## 4  LEARNED RL ALGORITHM RESULTS AND ANALYSIS

We discuss the training setup and results of our experiments. We highlight two learned algorithms with good generalization performance, DQNClipped and DQNReg, and analyze their structure.

### 4.1  TRAINING SETUP

***Meta-Training details:*** We search over programs with maximum 20 nodes, not including inputs or parameter nodes. A full list of node types is provided in Appendix A. We use a population size of 300, tournament size of 25, and choose these parameters based on the ones used in (Real et al., 2019). Mutations occur with probability 0.95. Otherwise a new random program is sampled. The search is done over 300 CPUs and run for roughly 72 hours, at which point around 20,000 programs have been evaluated. The search is distributed such that any free CPU is allocated to a proposed individual such that there are no idle CPUs. Further meta-training details are in Appendix B.

***Training environments:*** The choice of training environments greatly affects the learned algorithms and their generalization performance. At the same time, our training environments should be not too computationally expensive to run as we will be evaluating thousands of RL algorithms. We use

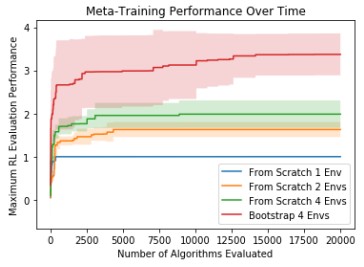
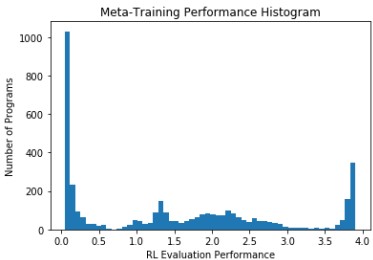

|(a) Learning curve|(b) Performance histogram|

Figure 3: Left: Meta-training performance over different number of environments from scratch, and boot-strapping. Plotted as RL evaluation performance (sum of normalized training return across the training environments) over the number of candidate algorithms. Shaded region represents one standard deviation over 10 random seeds. More training environments leads to better algorithms. Bootstrapping from DQN speeds up convergence and higher final performance. Right: Meta-training performance histogram for bootstrapped training. Many of the top programs have similar structure (Appendix D).

a range of 4 classical control tasks (CartPole, Acrobat, MountainCar, LunarLander) and a set of 12 multitask gridworld style environments from MiniGrid (Chevalier-Boisvert et al., 2018). These environments are computationally cheap to run but also chosen to cover a diverse set of situations. This includes dense and sparse reward, long time horizon, and tasks requiring solving a sequence of subgoals such as picking up a key and unlocking a door. More details are in Appendix C.

The training environments always include CartPole as an initial hurdle. If an algorithm succeeds on CartPole (normalized training performance greater than $0.6$), it then proceeds to a harder set of training environments. For our experiments, we choose these training environments by sampling a set of 3 environments and leave the rest as test environments. For learning from scratch we also compare the effect of number of training environments on the learned algorithm by comparing training on just CartPole versus training on CartPole and LunarLander.

***RL Training details:*** For training the RL agent, we use the same hyperparameters across all training and test environments except as noted. All neural networks are MLPs of size (256, 256) with ReLU activations. We use the Adam optimizer with a learning rate of $0.0001$. $\epsilon$ is decayed linearly from $1$ to $0.05$ over $1e3$ steps for the classical control tasks and over $1e5$ steps for the MiniGrid tasks.

## 4.2 Learning Convergence

Figure 3a shows convergence over several training configurations. We find that at the end of training roughly $70\%$ of proposed algorithms are functionally equivalent to a previously evaluated program, while early hurdles cut roughly another $40\%$ of proposed non-duplicate programs.

***Varying number of training environments:*** We compare learning from scratch with a single training environment (CartPole) versus with two training environments (CartPole and LunarLander). While both experiments reach the maximum performance on these environments (Figure 3a), the learned algorithms are different. The two-environment training setup learns the known TD loss

$$L_{DQN} = (Q(s_t, a_t) - (r_t + \gamma * \max_a Q_{targ}(s_t, a)))^2$$

while the single-environment training setup learns a slight variation $L = (Q(s_t, a_t) - (r_t + \max_a Q_{targ}(s_t, a)))^2$ that does not use the discount, indicating that the range of difficulty on the training environments is important for learning algorithms which can generalize.

***Learning from scratch versus bootstrapping:*** In Figure 3a, we compare training from scratch versus training from bootstrapping on four training environments (CartPole, KeyCorridorS3R1, DynamicObstacle-6x6, DoorKey-5x5). The training performance does not saturate, leaving room for improvement. Bootstrapping from DQN significantly improves both the convergence and performance of the meta-training, resulting in a $40\%$ increase in final training performance.

## 4.3 LEARNED RL ALGORITHMS: DQNCLIPPED AND DQNREG

In this section, we discuss two particularly interesting loss functions that were learned by our method, and that have good generalization performance on the test environments. Let

$$Y_t = r_t + \gamma * \max_a Q_{targ}(s_t, a), \text{ and } \delta = Q(s_t, a_t) - Y_t.$$

The first loss function DQNClipped is

$$L_{\text{DQNClipped}} = \max \left[ Q(s_t, a_t), \delta^2 + Y_t \right] + \max \left[ Q(s_t, a_t) - Y_t, \gamma (\max_a Q_{targ}(s_t, a))^2 \right].$$

$L_{\text{DQNClipped}}$ was trained from bootstrapping off DQN using three training environments (LunarLander, MiniGrid-Dynamic-Obstacles-5x5, MiniGrid-LavaGapS5). It outperforms DQN and double-DQN, DDQN, (van Hasselt et al., 2015) on both the training and unseen environments (Figure 4). The intuition behind this loss function is that, if the Q-values become too large (when $Q(s_t, a_t) > \delta^2 + Y_t$), the loss will act to minimize $Q(s_t, a_t)$ instead of the normal $\delta^2$ loss. Alternatively, we can view this condition as $\delta = Q(s_t, a_t) - Y_t > \delta^2$. This means when $\delta$ is small enough then $Q(s_t, a_t)$ are relatively close and the loss is just to minimize $Q(s_t, a_t)$.

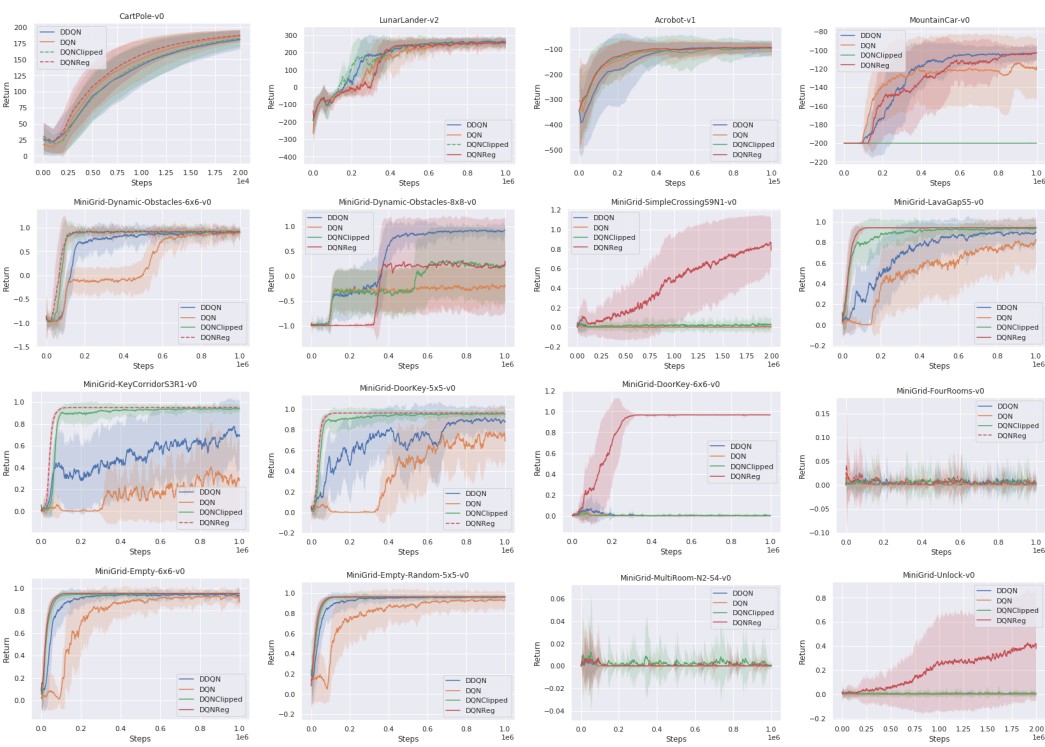

Figure 4: Performance of learned algorithms (DQNClipped and DQNReg) versus baselines (DQN and DDQN) on training and test environments as measured by episode return over 10 training seeds. A dashed line indicates that the algorithm was meta-trained on that environment while a solid line indicates a test environment. DQNReg can match or outperform the baselines on almost all the training and test environments. Shaded regions correspond to 1 standard deviation.

The second learned loss function, which we call DQNReg, is given by

$$L_{\text{DQNReg}} = 0.1 * Q(s_t, a_t) + \delta^2.$$

DQNReg was trained from bootstrapping off DQN using three training environments (KeyCorridorS3R1, Dynamic-Obstacles-6x6, DoorKey-5x5). In comparison to DQNClipped, DQNReg directly regularizes the Q values with a weighted term that is always active. We note that both of these loss functions modify the original DQN loss function to regularize the Q-values to be lower in value. While DQNReg is quite simple, it matches or outperforms the baselines on all training and test environments including from classical control and Minigrid. It does particularly well on a few test

environments (SimpleCrossingS9N1, DoorKey-6x6, and Unlock) and solves the tasks when other methods fail to attain any reward. It is also much more stable with lower variance between seeds, and more sample efficient on test environments (LavaGapS5, Empty-6x6, Empty-Random-5x5).

In Table 1, we evaluate DQNReg on a set of Atari games. We use the same architecture as in DQN (Mnih et al., 2013) and use the same no-op evaluation procedure which evaluates a trained policy every 1 million training steps over 200 test episodes. Even though meta-training was on computationally simple, non-image based environments, we find that DQNReg can generalize to image-based environments and outperform baselines. The results for the baselines are taken from their respective papers (Mnih et al., 2013; van Hasselt et al., 2015; Schulman et al., 2017).

| Env | DQN | DDQN | PPO | DQNReg |
|---|---|---|---|---|
| Asteroid | 1364.5 | 734.7 | 2097.5 | **2390.4** |
| Bowling | 50.4 | 68.1 | 40.1 | **80.5** |
| Boxing | 88.0 | 91.6 | 94.6 | **100.0** |
| RoadRunner | 39544.0 | 44127.0 | 35466.0 | **65516.0** |

Table 1: Performance of learned algorithm DQNReg against baselines on several Atari games. Baseline numbers taken from reported papers.

These algorithms are related to recently proposed RL algorithms, conservative Q-learning (CQL) (Kumar et al., 2020) and M-DQN (Vieillard et al., 2020). CQL learns a conservative Q-function by augmenting the standard Bellman error objective with a simple Q-value regularizer: $\log \sum_a \exp\left(Q(s_t, a)\right) - Q(s_t, a_t)$ which encourages the agent to stay close to the data distribution while maintaining a maximum entropy policy. DQNReg similarly augments the standard objective with a Q-value regularizer although does so in a different direction by preventing overestimation. M-DQN modifies DQN by adding the scaled log-policy (using the softmax Q-values) to the immediate reward. Both of these methods can be seen as ways to regularize a value-based policy. This resemblance indicates that our method can find useful structures automatically that are currently being explored manually, and could be used to propose new areas for researchers to explore.

We discover that the best performing algorithms from the experiment which learned DQNReg are consistent, and in the form $L = \delta^2 + k * Q(s_t, a_t)$. This loss could use further analysis and investigation, possibly environment-specific tuning of the parameter $k$. See Appendix 3 for details.

## 4.4 ANALYSIS OF LEARNED ALGORITHMS

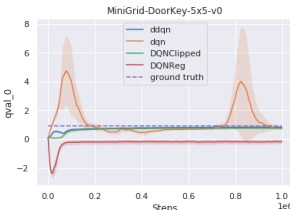 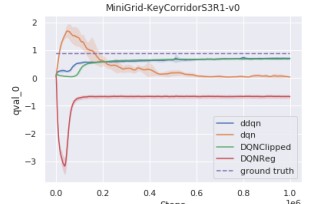 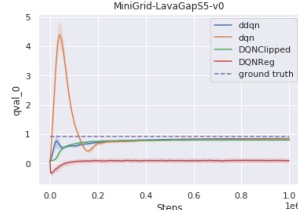

Figure 5: Overestimated value estimates is generally problematic in value-based RL. Our method learns algorithms which regularize the Q-values helping with overestimation. We compare the estimated Q-values for our learned algorithms and baselines with the optimal ground truth Q-values across several environments during training. Estimate is for taking action zero from the initial state of the environment. While DQN overestimates the Q-values, our learned algorithms DQNClipped and DQNReg underestimate the Q-values.

We analyze the learned algorithms to understand their beneficial effect on performance. In Figure 5, we compare the estimated Q-values for each algorithm. We see that DQN frequently overestimates the Q values while DDQN consistently underestimates the Q values before converging to the ground truth Q value which are computed with a manually designed optimal policy. DQNClipped has similar performance to DDQN, in that it also consistently underestimates the Q values and does so slightly more aggressively than DDQN. DQNReg significantly undershoots the Q values and does not converge to the ground truth. Various works (van Hasselt et al., 2015; Haarnoja et al., 2018; Fujimoto et al., 2018) have shown that overestimated value estimates is problematic and restricting the overestimation improves performance.

The loss function in DQNClipped is composed of the sum of two max operations, and so we can analyze when each update rule is active. We interpret DQNClipped as $\max(v_1, v_2) + \max(v_2, v_3)$ with four cases: 1) $v_1 > v_2$ and $v_3 > v_4$ 2) $v_1 > v_2$ and $v_3 < v_4$ 3) $v_1 < v_2$ and $v_3 < v_4$ 4) $v_1 < v_2$ and $v_3 > v_4$. Case 2 corresponds to minimizing the Q values. Case 3 would correspond to the normal DQN loss of $\delta^2$ since the parameters of $Q_{targ}$ are not updated during gradient descent.

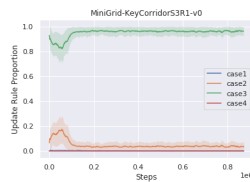

In Figure 6, we plot the proportion of when each case is active during training. We see that usually case 3 is generally the most active with a small dip in the beginning but then stays around $95\%$. Meanwhile, case 2, which regularizes the Q-values, has a small increase in the beginning and then decreases later, matching with our analysis in Figure 6, which shows that DQNClipped strongly underestimates the Q-values in the beginning of training. This can be seen as a constrained optimization where the amount of Q-value regularization is tuned accordingly. The regularization is stronger in the beginning of training when overestimation is problematic ($Q(s_t, a_t) > \delta^2 + Y_t$) and gets weaker as $\delta^2$ gets smaller.

Figure 6: Our learned algorithm, DQNClipped, can be broken down into four update rules where each rule is active under certain conditions. Case 3 corresponds to normal TD learning while case 2 corresponds to minimizing the Q-values. Case 2 is more active in the beginning when value overestimation is a problem and then becomes less active as it is no longer needed.

## 5 CONCLUSION

In this work, we have presented a method for learning reinforcement learning algorithms. We design a general language for representing algorithms which compute the loss function for value-based model-free RL agents to optimize. We highlight two learned algorithms which although relatively simple, can obtain good generalization performance over a wide range of environments. Our analysis of the learned algorithms sheds insight on their benefit as regularization terms which are similar to recently proposed algorithms. Our work is limited to discrete action and value-based RL algorithms that are close to DQN, but could easily be expanded to express more general RL algorithms such as actor-critic or policy gradient methods. How actions are sampled from the policy could also be part of the search space. The set of environments we use for both training and testing could also be expanded to include a more diverse set of problem types. We leave these problems for future work.

### ACKNOWLEDGEMENTS

We thank Luke Metz for helpful early discussions and feedback on the paper, Hanjun Dai for early discussions on related research ideas, and Xingyou Song, Krzysztof Choromanski, and Kevin Lee for help with infrastructure. We also thank Jongwook Choi for help with environment selection.

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

# A  SEARCH LANGUAGE DETAILS

Inputs and outputs to nodes in the computational graph have data types which include state $\mathbb{S}$, action $\mathbb{Z}$, float $\mathbb{R}$, list $List[\mathbb{X}]$, probability $\mathbb{P}$, vector $\mathbb{V}$. The symbol $\mathbb{X}$ indicates it can be of $\mathbb{S}, \mathbb{R}$, or $\mathbb{V}$. We assume that vectors are of fixed length 32 and actions are integers. Operations will broadcast so that for example adding a float variable to a state variable will result in the float being added to each element of the state. This typing allows the learned program to be domain agnostic. The full list of operators is listed below.

| Operation | Input Types | Output Type |
|---|---|---|
| Add | $\mathbb{X}, \mathbb{X}$ | $\mathbb{X}$ |
| Subtract | $\mathbb{X}, \mathbb{X}$ | $\mathbb{X}$ |
| Max | $\mathbb{X}, \mathbb{X}$ | $\mathbb{X}$ |
| Min | $\mathbb{X}, \mathbb{X}$ | $\mathbb{X}$ |
| DotProduct | $\mathbb{X}, \mathbb{X}$ | $\mathbb{R}$ |
| Div | $\mathbb{X}, \mathbb{X}$ | $\mathbb{X}$ |
| L2Distance | $\mathbb{X}, \mathbb{X}$ | $\mathbb{R}$ |
| MaxList | $List[\mathbb{R}]$ | $\mathbb{R}$ |
| MinList | $List[\mathbb{R}]$ | $\mathbb{R}$ |
| ArgMaxList | $List[\mathbb{R}]$ | $\mathbb{Z}$ |
| SelectList | $List[\mathbb{X}], \mathbb{Z}$ | $\mathbb{X}$ |
| MeanList | $List[\mathbb{X}]$ | $\mathbb{X}$ |
| VarianceList | $List[\mathbb{X}]$ | $\mathbb{X}$ |
| Log | $\mathbb{X}$ | $\mathbb{X}$ |
| Exp | $\mathbb{X}$ | $\mathbb{X}$ |
| Abs | $\mathbb{X}$ | $\mathbb{X}$ |
| (C)NN:$\mathbb{S} \rightarrow List[\mathbb{R}]$ | $\mathbb{S}$ | $List[\mathbb{R}]$ |
| (C)NN:$\mathbb{S} \rightarrow \mathbb{R}$ | $\mathbb{S}$ | $\mathbb{R}$ |
| (C)NN:$\mathbb{S} \rightarrow \mathbb{V}$ | $\mathbb{V}$ | $\mathbb{V}$ |
| Softmax | $List[\mathbb{R}]$ | $\mathbb{P}$ |
| KLDiv | $\mathbb{P}, \mathbb{P}$ | $\mathbb{R}$ |
| Entropy | $\mathbb{P}$ | $\mathbb{R}$ |
| Constant | | 1, 0.5, 0.2, 0.1, 0.01 |
| MultiplyTenth | $\mathbb{X}$ | $\mathbb{X}$ |
| Normal(0, 1) | | $\mathbb{R}$ |
| Uniform(0, 1) | | $\mathbb{R}$ |

# B  TRAINING DETAILS

We describe the training details and hyperparameters used. For all environments we use the Adam optimzier with a learning rate of 0.0001.

**Common RL training details.** All neural networks are MLPs of size (256, 256) with ReLU activations. For optimizing the Q-function parameters we use the Adam optimizer with a learning rate of 0.0001. Target update period is 100. These settings are used for all training and test environments.

**Classical control environments.** The value of $\epsilon$ is decayed linearly from 1 to 0.05 over 1000 steps. CartPole, Acrobat, and MountainCar are trained for 400 episodes and LunarLander is trained for 1000 episodes.

**MiniGrid environments.** The value of $\epsilon$ is decayed linearly from 1 to 0.05 over $10^5$ steps. During meta-training, MiniGrid environments are trained for $5 * 10^5$ steps.

**Atari environments.** We use the same neural network architecture as in Mnih et al. (2013). Target update period is $1,000$. The value of $\epsilon$ is decayed linearly from 1 to 0.1 over $10^6$ steps. For evaluation, we use the no-op start condition as in Mnih et al. (2013) where the agent will output the no-op action for $x$ steps where $x$ is a random integer drawn between $[1, 30]$. The evaluation policy uses an $\epsilon$ of 0.001 and is evaluated every $10^6$ steps for 100 episodes. The best training snapshot is reported.

# C  ENVIRONMENT DETAILS

We describe the classical control environments below. CartPole and LunarLander are dense reward while Acrobat and MountainCar are sparse reward.

| | Task ID | Description |
|---|---|---|
|  | CartPole-v0 | The agent must balance a pole on top of a cart by applying a force of $+1$ or $-1$ to the cart. A reward of $+1$ is provided for each timestep the pole remains upright. |
|  | LunarLander-v2 | The agent controls a lander by firing one of four thrusters and must land it on the landing pad. |
|  | Acrobat-v1 | The goal is to swing a 2-link system upright to a given height by applying 1, 0, or -1 torque on the join between the two links. |
|  | MountainCar-v0 | The goal is to drive up the mountain on the right by first driving back and forth to build up momentum. |

We describe the MiniGrid environments below. The input to the agent is a fully observed grid which is encoded as an NxNx3 size array where N is the grid size. The 1st channel contains the index of the object type at that location (out of 11 possible objects), the 2nd channel contains the color of the object (out of 6 possible colors), and the 3rd channel contains the orientation of the agent out of 4 cardinal directions. This encoding is then flattened and fed into an MLP. There are 7 possible actions (turn left, turn right, forward, pickup, drop, toggle, done).

Unless stated otherwise, all tasks are sparse reward tasks with a reward of 1 for completing the task. Max steps is set to 100. A size such as 5x5 in the environment name refers to a grid size with width and height of 5 cells.

| | Task ID | Description |
|---|---|---|
|  | KeyCorridorS3R1-v0 | The agent has to find a key hidden in one room and then use it to pickup an object behind a locked door in another room. This tests sequential subgoal completion. |
|  | LavaGapS5-v0 | The agent has to reach the green goal square without touching the lava which will terminate the episode with zero reward. This tests safety and safe exploration. |
|  | MultiRoom-N2-S4-v0 | The agent must open a door to get to the green goal square in the next room. |

| | | |
|---|---|---|
|  | SimpleCrossingS9N1-v0 | The agent has to reach the green goal square on the other corner of the room and navigate around walls. |
|  | Empty-v0 | The agent has to reach the green goal square in an empty room. |
|  | EmptyRandom-v0 | The agent has to reach the green goal square in an empty room but is initialized to a random location. |
|  | Dynamic-Obstacles-v0 | The agent has to reach the green goal square without colliding with any blue obstacles which move around randomly. If the agent collides with an obstacle it receives a reward of $-1$ and the episode terminates. |
|  | FourRooms-v0 | The agent must navigate in a maze composed of four rooms. Both the agent and goal square are randomly placed in any of the four rooms. |
|  | DoorKey-v0 | The agent must pick up a key to unlock a door to enter another room and get to the green goal square. |

## D  GRAPH DISTRIBUTION ANALYSIS

We look at the distribution of top performing graphs and find similarities in their structure. This is summarized in Table 3 where we describe the equations of learned algorithms for differing ranks (if sorted by score). The best performing algorithms from the experiment which learned DQNReg are all variants of adding $Q(s_t, a_t)$ to the standard TD loss in some form, $\delta^2 + k * Q(s_t, a_t)$. We think this kind of loss could use further investigation and that while we did not tune the value of $k$, this could also be tuned per environment. In Figure 3b, we show the distribution of scores for all non-duplicate programs that have been evaluated. We provide a full list of top performing algorithms from a few of our experiments at `https://github.com/jcoreyes/evolvingrl`.

| Raw Equation | Simplified Equation | Score | Rank |
|---|---|---|---|
| $\delta^2 + 0.1 * Q(s_t, a_t) + r_t - (\gamma * Q_{targ} - 0.1 * Q(s_t, a_t))$ | $\delta^2 + 0.2 * Q(s_t, a_t)$ | 3.905 | 2 |
| $\delta^2 + 0.1 * Q(s_t, a_t) - \gamma + Q_{targ}$ | $\delta^2 + 0.1 * Q(s_t, a_t)$ | 3.904 | 3 |
| $\delta^2 - (\gamma * Q_{targ} - 0.1 * Q(s_t, a_t))$ | $\delta^2 + 0.1 * Q(s_t, a_t)$ | 3.903 | 4 |
| $\delta^2 + Q_{targ} + 0.1 * Q(s_t, a_t) - \gamma$ | $\delta^2 + 0.1 * Q(s_t, a_t)$ | 3.902 | 5 |
| $\delta^2 - (0.1 * Q(s_t, a_t) - Y_t)^2$ | $\delta^2 - (0.1 * Q(s_t, a_t) - Y_t)^2$ | 3.898 | 6 |
| $\delta^2 + ((r_t + \gamma * Q_{targ} + Q(s_t, a_t)) * (\gamma - \max(\gamma, 0.1 * Q(s_t, a_t))$ $-\gamma * Q_{targ} - 0.1 * Q(s_t, a_t))$ | NA | 3.846 | 11146 |
| $\delta^2 + (\delta^2 + 0.1 * Q(s_t, a_t))^2$ | NA | 3.65 | 12146 |
| $\delta^2 + Q(s_t, a_t)$ | $\delta^2 + Q(s_t, a_t)$ | 2.8 | 12446 |
| $\delta^2$ | $\delta^2$ | 2.28 | 13246 |

Table 3: Other programs learned in learning DQNReg which is rank 1 with score 3.907. Rank is if scores are sorted in decreasing order. Score is the sum of normalized RL training performance across four environments. The simplified equations contains only the relevant parts for minimizing the equation output. $Q_{targ}$ refers to $\max_a Q_{targ}(s_{t+1}, a)$.

## E  REPEATABILITY OF META TRAINING

In Figure 7, we plot the meta-training performance for bootstrapping from DQN with four training environments (CartPole, KeyCorridorS3R1, Dynamic-Obstacles-6x6, DoorKey-5x5) over ten trials. Four out of the ten trials reach the max training performance. Two out of 10 of these trials learns the same algorithm DQNReg while the other top two trials find other less interpretable algorithms.

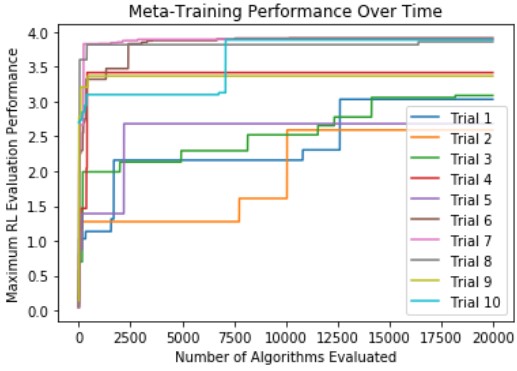

Figure 7: Meta-training performance for boot-strapping on 4 training environments for 10 random seeds.

