# OpenReview forum: "Evolving Reinforcement Learning Algorithms"
_ICLR.cc/2021/Conference — ICLR 2021 Oral_

### Official Review · AnonReviewer2 · 2020-10-26
**A bit old-fashioned idea, but checks outs**

**Rating:** 9
**Confidence:** 4

**Review:**

The paper proposes an approach to develop new Reinforcement Learning (RL) algorithms through a population-based method very reminiscent of Genetic Programming (GP). The authors "evolve" loss functions that can be used across different RL scenarios and providing good generalization.

Pros:
- The paper is well written, well structured and clear at all times. There are few typos here and there but nothing that affects the readability of the paper.
- The results are relevant: the fact that the method re-discovers recently proposed learning models is remarkable, as well as it finds new models indicates that it is a line of research worthy of further exploration.
- The other minor contributions are also noteworthy, specifically the Functional equivalence check, which IMO is a brilliant idea, as well as the early hurdles approach.
- The interpretation of the approach is also correct: the fact that authors make a clear distinction between learning from scratch and bootstrapping tells me that they truly understand the overall framework they based their method on.

Cons:
- The main idea behind this type of meta-learning is always very interesting to revisit. Nevertheless, and truth to be told, the idea of using GP to find new loss (or "objective/fitness", as known in evolutionary computation (EC)) functions is quite old [1,2]. At some points the work presented here feels somewhat "old-fashioned", or a mere re-edition or adaptation of those early seminal works.
- The results can be easily rebutted; if I understand correctly, the authors are presenting the results of a SINGLE run for every scenario they tested their method on (different no. of environments) , which tell us nothing about the average behavior of their proposed approach. Although this fact is understandable given such long training times (3 days with 300 cpus for a single run), many people will not accept the presented results arguing that they could have been the result of lucky runs. On the bright side, such practice is somewhat usual in the deep learning community, so many people may overlook it for now. I'd suggest the authors make a statement arguing why they feel confident that they approach may present a low variance, or why they still consider these results relevant, even if they require twenty or 30 runs to find results this good. They could argue that even if their method is currently statistically unreliable, it could probably be stabilized with many methods proposed in the EC community (such as spatially distributed populations [3], etc.)
- The thing that bugs me the most of this work is that the proposed method is clearly a flavor of GP; however the authors never really acknowledge it by its name; instead they claim it to be something called "Regularized Evolution" which is supposedly introduced in a previous paper; nevertheless, it is really just GP: they are evolving tree-shaped programs, which is the hallmark of GP (there exist variants of GP that are not even EC-based [4]), so I don't see a reason for not calling it for what it is. I wish the authors clearly state why their approach cannot be considered part of the GP framework, or if it indeed is, then make such statement clearly.
- In this vein, it is also interesting to note that they do not use a crossover operation. In the GP framework, crossover is generally regarded as a more powerful operator than mutation, and the combination of both operators can give the best results. My guess is that crossover is difficult to implement given the "search language" (or 'primitives', as known in GP), and then crossover would result in many invalid programs. Still, it would be nice if authors could explain a little bit on this issue.

Other comments:
It draw to my attention that you use populations of size 300 as well as a 300 cpu system. It would seem that you intend to have one cpu for each individual evaluation; however, given the advances presented by the functional equivalence check and the early hurdles technique, I wonder what happens when a individual is no longer being evaluated.. is that cpu left idle? or is it reassigned to contribute to the evaluation of another individual (which sound complex to do, given the RL nature of the problems being treated)?

References:
1. Bengio, S., Bengio, Y., & Cloutier, J. (1994, June). Use of genetic programming for the search of a new learning rule for neural networks. In Proceedings of the First IEEE Conference on Evolutionary Computation. IEEE World Congress on Computational Intelligence (pp. 324-327). IEEE.
2. Trujillo, L., & Olague, G. (2006, July). Synthesis of interest point detectors through genetic programming. In Proceedings of the 8th annual conference on Genetic and evolutionary computation (pp. 887-894).
3. Tomassini, M. (2006). Spatially structured evolutionary algorithms: Artificial evolution in space and time. Springer Science & Business Media.
4. Hooper, D. C., Flann, N. S., & Fuller, S. R. (1997). Recombinative hill-climbing: A stronger search method for genetic programming. Genetic Programming, 174-179.

---

> ### Author Response · Authors · 2020-11-16
> **R2 Response**
>
> Thank you for the detailed and helpful feedback. We are glad the reviewer found the results relevant and noteworthy.
>
> We agree that Regularized Evolution falls under GP and that we could have done more to reference it. We have revised the paper to more explicitly state that our approach falls under GP in the introduction and Section 3.3, and revised the related work to include the references you have highlighted. You are correct in that we decided to not use the crossover operation because it would lead to many invalid programs and is difficult to implement.
>
> The meta-training runs can have high variance as these methods do not always converge to a good solution. For the learned algorithms, we ran the same configuration less than five times. As you have mentioned, this could be stabilized with various techniques. We also think that even if the process is high variance, at the end, we show we can still learn a generalizable algorithm which transfers to a wide variety of environments. Furthermore, we plan on releasing the full list of learned algorithms and their performance and think this could be a useful database for other researchers to analyze and learn from.
>
> Regarding the population and CPU system size, the population and number of cpus does not have to be the same. Once a CPU has evaluated an individual, it is reassigned to contribute to the evaluation of another individual. There is a chief node which will propose new algorithms for any idle CPUs. We have added this information to the paper in Section 4.1.

---

### Official Review · AnonReviewer3 · 2020-10-28
**Interesting Work**

**Rating:** 6
**Confidence:** 3

**Review:**

##########################################################################

Summary:

The paper introduces learning of RL algorithms via evolution. Over here, RL algorithms are represented as computational graphs  where the graph outputs the objective of the algorithm to be minimized. Also, the graph  comprises pre-defined symbols and operations.
Given these computational graphs or RL algorithms, they use Regularized Evolution(RE) to evolve a RL algo. For evolution, each RL algo is evaluated with a set of training envs. Also, there are various checks introduced to avoid incorrect algos or re-evaluation of the same algo. For the mutation, the algo which performs best across all training envs is selected and it’s mutations are reintroduced in the population.
These computational graphs can be initialized randomly or existing RL algos could be induced into them for bootstrapping.

##########################################################################

Reasons for score:

I vote for marginal acceptance. I believe the notions introduced in this work could be useful for the community and their results seem to indicate a promising direction.

##########################################################################

Pros:

- They learn two new RL algos  “DQN clipped” &  “DQNreg”  which happen to be better  sample efficient  and have better final  performance than DQN. As per authors, these algorithms also share resemblance to recent CQL and M-DQN.
- Promise of better algorithms than researchers can design given the coverage of all reasonable operators/symbols in the computational graphs.
- They performed ablation study of with/ without bootstrapping of RL algorithms.

#########################################################

Cons:
- Large number of CPUs required.

#########################################################

Questions:

- How to make a choice of environments for meta-training?
- Why are the authors using training return for performance of the Algorithm?  After training for M episodes, they can simply evaluate the greedy policy for “k” episodes and that metric should be used for performance measure. Otherwise, we are using an epsilon-greedy metric for comparison which could be really low as that’s dependent on epsilon and environment.
- It’s not clear if they are using constant epsilon or decaying epsilon?

Other comments:

Section 4.2, 2nd para:
“The two-environment training setup is learns the known … “ => remove “is”

---

> ### Author Response · Authors · 2020-11-16
> **R3 Response**
>
> Thank you for the helpful feedback. We are glad the reviewer found the results useful for the community. Regarding the larger number of CPUs required, our experiments can be run with less CPUs but will take longer to run. The number of CPUs can be less than the population size.
>
> The choice of training environments is quite important and can be difficult to choose. The training environments should ideally be computationally cheap yet cover a diverse range of scenarios. Therefore, we try to choose training environments from the MiniGrid suite that cover different scenarios from sparse reward (DoorKey), safety (DynamicObstacle), and sequential subgoals (KeyCorridor). CartPole is also used as a hurdle environment to quickly weed out poor performing algorithms.
>
> We first tried to use evaluation return but then we decided to use training return because this also allows us to optimize for algorithm sample efficiency as well. We have revised the paper to explain this justification in Section 3.1. If we used just evaluation return of the greedy policy at the end of training, then our search would skip over an algorithm that has similar final evaluation performance but is more sample efficient. This is observed in some of the environments such as MiniGridEmpty-6x6 (Figure 4), where the final performance is similar but DQNReg is more sample efficient. Since we are using a decaying epsilon (described in Appendix B), training return towards the end of training will be close to the evaluation return of a greedy policy.

---

### Official Review · AnonReviewer4 · 2020-10-28
**Impressive approach to meta-learning RL algorithms with interesting analysis**

**Rating:** 7
**Confidence:** 3

**Review:**

This paper proposes a method of discovering, through evolutionary search, programatically defined RL algorithms.

Pros:


-- Novel (to my knowledge) in the search language it uses, which renders the resulting learned algorithms very interpretable and generalizable (as compared to, say, using a meta-learned loss function parameterized by a neural network)

-- Interesting analyses of particular meta-learned algorithms

-- Useful discussion of key implementation details (e.g. early hurdles, functional equivalence check)

Cons

-- It would be nice to see more analysis of the algorithms obtained "from scratch" rather than starting from DQN.   Are any other (variants of) standard RL algorithms discovered?  Are there any new but interpretable algorithms that perform well?

-- Comparison to other approaches to meta-learned loss functions (e.g. Kirsch et al., or Oh et al.) would be helpful

-- I realize it may be computationally infeasible to meta-train on many more environments, or more difficult tasks, but it would at least be good to see how the learned algorithms generalize to environments considerably unlike any seen during meta-training (e.g. not gridworld and not "classical control" -- perhaps Atari?)

---

> ### Author Response · Authors · 2020-11-16
> **New Results on Atari**
>
> Thank you for the helpful feedback and comments. We are glad the reviewer found the learned algorithms to be interpretable and generalizable. We have added new results on Atari games, showing that DQNReg outperforms baselines on all the Atari games that we tested. Table 1 summarizes the results. The Atari environments are image-based and unlike any seen during meta-training, indicating that the learned algorithms can generalize to completely different environments.
>
> Regarding analysis of algorithms learned from scratch, we found that our method is able to learn TD error from scratch, and learns other variants such as TD without the discount but these variants do not perform well across the environments. Unsurprisingly,  performance of learning from scratch to be subpar in general because learning from scratch is more difficult and the language selection is limited to value based methods. This might suggest that more work can be done to improve the meta-learning optimization and expand the search space to prevent the algorithm from getting stuck in local minima. We plan on releasing the full list of learned algorithms and their performance and think this could be a useful database for other researchers to analyze and learn from.
>
> We agree with the reviewer that it would be helpful to compare to other meta-learned loss functions (Kirsh et al. or Oh et al.). We did not have time to run these comparisons and will run these in the future. We highlight that these meta-learned loss functions are continuously parameterized with a LSTM and thus are not interpretable compared to our method.

---

### Author Response · Authors · 2020-11-23
**Summary of Changes**

We thank the reviewers for their time and helpful feedback. We summarize the changes we made in response to the reviewers’ feedback, including newly added dataset with learning algorithms:

-- We have released the data for the top 500 algorithms for both learning from scratch and learning from bootstrapping experiments to enable further theoretical analysis of the discovered algorithms and justify the computational cost of obtaining them. This data contains the score for the algorithm and an image of the computational graph. The data contained in the supplementary zip file along with a README for how to parse it.

-- We have added results on Atari games, showing that one of the learned algorithms, DQNReg, outperforms baselines on all the Atari games that we tested. Table 1 summarizes the results.

-- We have revised the writing to acknowledge that Regularized Evolution falls under GP.

-- We have updated Table 3 in the appendix with more loss functions. The distribution of loss functions show that the top found algorithms are consistent with each other.

-- We have updated the writing to address various questions including the use of training return to score the algorithms and how CPUs are allocated during training.

---

### Decision · Program_Chairs · 2021-01-07
**Final Decision**

**Decision:**

Accept (Oral)

**Comment:**

This paper proposes a meta-learning algorithm for reinforcement learning. The work is very interesting for the RL community, it is clear and well-organized. The work is impressive and it contributes to the state-of-the-art.